# Simulation for Discrete Elements of the Powder Laying System in Laser Powder Bed Fusion

**Yini Song** [1,*], **Jun Wang** [2,*] , **Guangyu Lou** [2] **and Kai Zhang** [2]

1   National Engineering Research Center for Digital Manufacturing Equipment, Huazhong University of Science and Technology, Wuhan 430074, China
2   School of Mechanical Engineering, Hubei University of Technology, Wuhan 430068, China
*   Correspondence: songyini98@163.com (Y.S.); junwang@hbut.edu.cn (J.W.)

**Abstract:** For analyzing the influence of the system parameters on the density of the powder layers in laser powder bed fusion (LPBF) technology, an experimental method is proposed to improve the structure of the recoater in the powder laying system and optimize the parameters of the powder laying system. With this experimental method, the appropriate density of the powder layers can be attained. In the proposed experimental method, the recoater in the powder laying system was taken as the research object and the forces affecting the powder and recoater when the powder was in contact with the recoater were analyzed. The discrete element model of the powder laying system was established to simulate and analyze the influences of the recoater's radius, translational velocity and angular velocity on the density of powder layers. In addition, orthogonal experiments were designed to discuss the magnitude of the influence of each of the powder laying system's parameters on the density of powder layers. Finally, the optimized parameter combination plan was put forward. The results show that increasing the recoater's radius can enhance the density of powder layers within a certain range; but, as the recoater's radius is increased continuously, its impact on the recoater's radius on f powder layers' density decreases. When the translational velocity of the recoater rises, powder layers' density increases first and then decreases. The coater's angular velocity has little effect on powder layers' density. Eventually, the optimized processing parameters were determined, which are 25 mm for the recoater's radius, 30 mm/s for the recoater's translational velocity, and 12 s$^{-1}$ for the recoater's angular velocity. The results provide some significance and guidance in improving the recoater's structure and optimizing the powder laying system's parameters.

**Keywords:** laser powder bed fusion; powder laying system; discrete elements; parameter optimization





## 1. Introduction

Laser powder bed fusion (LPBF) is a laser additive manufacturing method with a laser as the heat source and metal powder as raw material, which rapidly produces parts and uses layered superposition technology to form a single layer by overlapping a single-melting path. Complex geometric components can be produced in a short period of time after layers are stacked. LPBF is of great significance for the personalized and rapid manufacture of metal parts [1–3]. However, in LPBF technology, there is always a problem of the shrinkage deformation of molded parts, which is not only related to the selection of processing parameters during molding, but is also related to the density of powder layers in the molding cylinder. Generally speaking, the density of the molded part is more than 98% of the full density, and the powder density in the molding cylinder is about 70% of the full density [4]. Therefore, the molded part shrinks and deforms due to the change of powder layers' density during molding. Higher powder layers' density can effectively reduce the gradient of density change in molding, thereby improving the performance of molded parts. Therefore, in LPBF technology, ensuring a high density of powder layer is one of the necessary prerequisites for sintering molded parts with excellent performance. Recently, a

large number of studies were carried out on the powder laying system for LPBF. Han [5] investigated the effect of powder layer thickness on the properties of various powder bed characteristics during single- and multi-layer powder deposition. Chen [6] used the discrete element method to study the deposition mechanism at the particle scale, including particle contact stress and particle velocity, and experimentally verified the bulk density at the macro scale. Zheng [7] established a powder microscopic contact force model under the consideration of the van der Waals force between particle molecules. Thus, the powder laying process simulation was carried out, and the influence of the microscopic parameters of the powder particles and the movement parameters of the powder recoater on the density of the powder layer was obtained. Zhang [8] established a discrete unit model of powder laying in additive manufacturing, and systematically analyzed the influence of the vibration of the powder recoater on the quality of the powder layer, indicating that the tiny vibration of the drum can effectively reduce the adhesion of the powder to the drum and improve the compactness of the powder bed. Parteli [9] studied the effect of powder diffusion conditions on the roughness of the powder layer, showing that the roughness of the powder layer increased as the translation velocity of the recoater increased. After studying the kinetics of powder diffusion under the powder recoater, the effect of the recoater pressure gap height and angular velocity on the uniformity of the powder layer was analyzed by discrete unit method simulation [10], which showed that the uniformity of the powder layer was improved with the increase in the recoater pressure gap height and the decrease in the angular velocity. Yang, P. et al. [11] prepared the pure Cu with high relative density and high strength by the LPBF technology with a surface oxidation treatment. Ma, C. et al. [12] created a single-layer three-dimensional model to simulate multi-channel scanning of AlSi$_{25}$ powder in laser powder bed fusion (LPBF) by the finite element method. Cosma, C. et al. [13] investigated the use of laser powder bed fusion (LPBF) in depositing a superior material, such as CoCr, on an existing stainless-steel base. In summary, the microscopic morphology of powder particles in the powder laying system, or the influence of a single variable on the quality of the powder layer was discussed. It fails to optimize the reasonable selection of the process parameters of the powder laying. In view of this, this paper takes the powder recoater as the research object, analyzes the force state of the powder when it is in contact with the powder recoater, and establishes a discrete unit model of the powder laying system in laser powder bed fusion. Under the premise of the influence law of each powder laying motion parameter (radius of the powder recoater, translational velocity, angular velocity) obtained by single factor analysis, the appropriate horizontal factor is selected for orthogonal testing, and the influence of each powder laying parameter on the density of the powder layer is discussed; finally, the optimized parameter combination scheme is proposed.

## 2. Analysis of Powder Spreading Motion Parameters

### 2.1. Model of the Powder in Contact with the Recoater

Figure 1 shows a schematic diagram of the contact between the recoater, scrapper and the powder, and the distribution of the powder state during the molding process can be roughly divided into three regions: I.—loose area, where the powder is only subject to scraper action from the scraper, natural accumulation and continuous movement along the moving direction; II.—deformation zone, where the powder comes into contact with the recoater, and the internal arch bridge effect is destroyed and the density changes; III.—molding area, where the loosely arranged powder is compacted and transformed into a paving layer with a certain density. When the metal powder transitions from zone I to zone II, the scraper pre-compacts the metal powder and reduces the height of the powder buildup before laying the powder stick, in which the splash metal caused by the spheroidization phenomenon in the molding and avoids the occurrence of stagnant rolling can be removed [14]. In the process of metal powder being converted from zone II to zone III, the shear force generated by the powder recoater will change the arrangement inside the powder layer, thereby realizing the compression of the volume of the powder layer.

However, due to the principle of slope self-locking, only part of the accumulated powder will be compacted by the powder recoater into the molding area, while the other part will be panned with the powder recoater. Theoretically, if the dividing point where the powder is translated or compacted occurs at zero force on the powder in the vertical direction, the force of the powder in the deformation zone and the height of the compaction are shown in Figure 2.

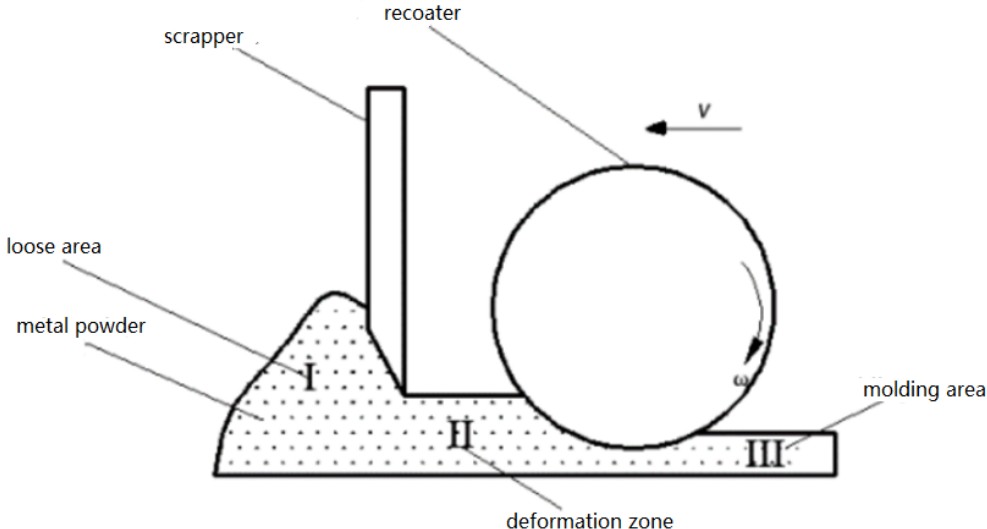

**Figure 1.** A motion model of the powder in contact with the recoater.

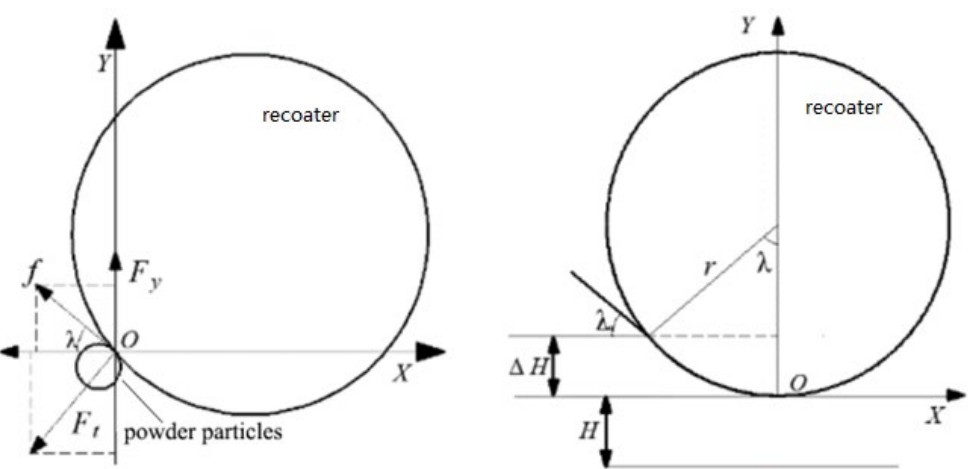

(**a**) Force of the powder particles        (**b**) The height of the powder being compacted

**Figure 2.** The force of the powder and the height of the compaction in the deformation zone.

Figure 2a shows the force analysis diagram of the powder in the deformation region, which treats the powder particles as a sphere, and establishes a Cartesian coordinate system with the tangent o as the coordinate origin. Then, the compound $F_y$ of the powder in the vertical direction is:

$$F_y = f \sin \lambda - F_t \cos \lambda \tag{1}$$

where $f$ is the friction force of the powder particles and $F_t$ is the positive pressure of the powder particles. Since the dividing point where the powder is flattened or compacted when the vertical direction of the combined force $F_y = 0$ is the dividing point, the angle between the tangent line between the powder particles and the laying stick at this time is

the critical angle $\lambda$, generated by the self-locking phenomenon, assuming that the static friction coefficient between the powder layer and the powder recoater is $\mu$, it can be seen:

$$\lambda = \arctan\frac{F}{f} = \arctan\frac{F}{\mu F} = \arctan\frac{1}{\mu} \tag{2}$$

Figure 2b shows a schematic diagram of the height of the powder being compacted in the deformation region, from which it can be seen that the height of the powder being compacted by force $\Delta H = r - r\cos\lambda$, assuming that the thickness of the coating layer is $H$, the compaction effect of the powder recoater on the powder layer can be seen as compressing the powder layer of $\Delta H + H$ height to a height of $H$, according to the mass conservation theorem, the percentage of its density increase is:

$$\eta = \frac{\rho_1 \Delta H}{\rho_2 H} = \frac{\rho_1 r(1 - \cos\lambda)}{\rho_2 H} \tag{3}$$

wherein, $\rho_1$ is the density of the powder in the deformation zone, and $\rho_2$ is the density of the powder in the molding zone. It can be seen from Equation (3) that the density of the powder layer after compaction by the powder recoater is positively correlated with the radius $r$ of the powder recoater and the critical angle $\lambda$. Consequently, in the actual equipment, in order to satisfy the density requirements of the powder laying, the larger diameter of the powder recoater is preferred, and because the critical angle $\lambda = \arctan 1/\mu$, in order to increase the critical angle, the coefficient of static friction between the powder recoater and the powder should be reduced $\mu$: that is, the surface of the powder recoater should be as smooth as possible.

### 2.2. Powder Layer Motion Model

The above force analysis is based on the moment when the powder particles and the powder recoater are in contact, and only the force of a single powder particle is considered. However, in the actual powder laying process, the movement of the powder layer is actually the result of the simultaneous movement of multiple powder particles, as shown in Figure 3, due to the gap between the powder particles. Then, in the process of the powder spreading movement, the powder particles with a smaller particle size may fall into the gap between the particles with a larger particle size, making the powder layer more dense and flat. Therefore, in order to reduce the porosity of the particle accumulation of the powder laying layer, in addition to the translation of the powder recoater, the rotational motion in the same direction is usually added to change the motion state of the powder layer, and the ideal transmission model is shown in Figure 4. In this way, a small number of particles in direct contact with the powder recoater will roll, so that the displacement distance of the surface particles in the horizontal direction increases, and the more likely the gap between the particles is to be filled.

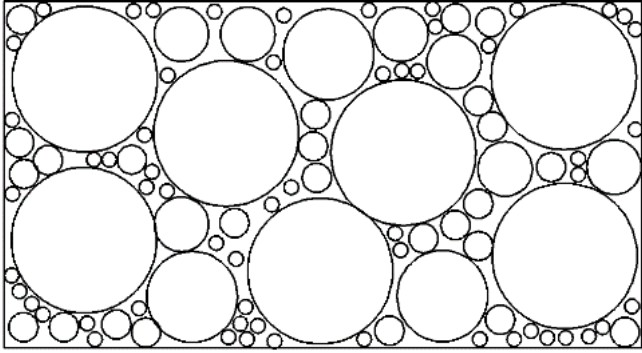

**Figure 3.** Pattern of accumulation of powder particles.

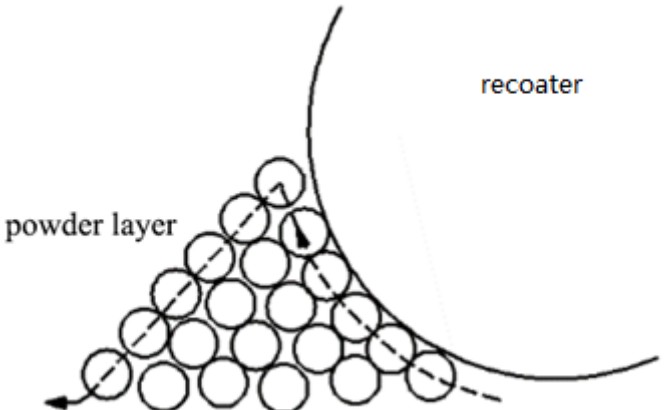

**Figure 4.** Ideal transport model for powder particles.

The increase in the displacement distance of the powder particles in the laying powder is actually a manifestation of the increased fluidity in the powder layer. From a macroscopic point of view, the main reason for this is that the powder layer is subjected to the force generated in the diagonal direction due to the rotation of the powder recoater, resulting in the movement mode of the powder particles from pure flat to flat and rolling, so that the obstruction of friction during the transmission process is reduced. Although the rotation of the powder recoater can improve the fluidity of the powder, its value should not be too large, otherwise it will make the dust fly in the powder laying, causing pollution and damage to the core components of the equipment. Similarly, the translational velocity of the powder recoater affects the force of the powder layer in the horizontal direction, and the change in its velocity will inevitably change the arrangement of the powder particles. Therefore, in order to obtain a high-quality coating layer, the motion parameters of the powder laying device should be comprehensively considered.

## 3. Simulation of the Spreading Motion Process

### 3.1. The Basic Theory of Discrete Element Simulation

The discrete element method (DEM) is a numerical calculation method proposed by Professor Cundall in 1971 [15,16]. The basic idea is to divide the research object into multiple relatively independent units, so that each element satisfies the equation of motion and then, according to the contact model and Newton's second law, the cycle iterative calculation of each element is determined to determine the force and displacement of all elements within the unit time step. Finally, the microscopic motion of each unit is tracked and calculated, reflecting the law of macroscopic motion of the study object.

### 3.2. The Simulation Model of Contact among Particles

Considering that the particle size of the actual powder is small, it is susceptible to the van der Waals force during the spreading process. Therefore, the Hertz–Mindlin JKR model was selected as the simulation contact model, which introduced the calculation of adhesion force based on Hertz elastic contact theory, which is suitable for the simulation of fine particles that need to consider the van der Waals force.

Based on the Hertz–Mindlin elastic model, the contact force model of particle i:

$$F_{CN} = F_{CN,S} + F_{CN,D} \tag{4}$$

In Equation (4), $F_{CN,S}$ is the normal contact force and $F_{CN,D}$ is the normal damping force.

$$\begin{cases} F_{CN,S} = -K_n \cdot \alpha_n^{1.5} \\ K_n = \frac{4 \cdot G \sqrt{r^*}}{3 \cdot (1-v)} \end{cases} \tag{5}$$

In Equation (5), $K_n$ is the Hertz contact constant, $\alpha_n$ is the normal contact displacement, $G$ is the shear modulus of the particle, $r^*$ is the radius of local curvature of the particle, and $v$ is the Poisson's ratio of the particle.

$$F_{CN,D} = -2\beta v_n^{rel} \sqrt{5S_n m^*/6} \tag{6}$$

In Equation (6), $m^*$ is the equivalent mass of the particles; $\beta$ is the damping coefficient; $S_n$ is the normal contact stiffness; $v$ is the normal relative velocity of the contact particles.

Calculation of the van der Waals force contact model based on Hamaker theory is as follows:

$$m_i \frac{dv_i}{dt} = \sum_j F_{ij}^c + \sum_k F_v + F_i^g \tag{7}$$

$$I_i = \frac{dw_i}{dt} = \sum_j M_{ij} \tag{8}$$

In Equations (7) and (8), $v_i$ and $w_i$ are particle $i$ translational and rotational velocity, respectively. $F_{ij}^c$, $M_{ij}$ is the contact force and torque between particles $i$ and $j$, or between particles and cylinder wall. $F_i^g$ is the gravitational force of particle $i$. $F_v$ is the van der Waals force. Based on the principle of gravitational potential energy and energy superposition of London–van der Waals, the van der Waals force is obtained.

$$F_v = -\frac{A}{12Z_0^2} \frac{d_1 d_2}{d_1 + d_2} \tag{9}$$

In Equation (9), $d_1$ and $d_2$ are the diameters of the two particles. $Z_0$ is the distance between particles. $A$ is the Hamaker constant. When there is no direct contact between particles, the particles are only affected by the van der Waals force. When the particles are in direct contact, the particles are affected by the contact force and the van der Waals force, and the van der Waals force is a fixed value.

## 4. Establishment of the Powder Model

This simulation adopts the internationally accepted discrete element modeling software EDEM (version number 2.7, created by DEM Solutions Ltd., sourced from Edinburgh, The United Kingdom), which can establish a parametric model of the particle solid system and simulate and analyze the formation and motion process of particles [17]. EDEM is commonly used in particle processing and analysis in industrial production. When EDEM software is used for powder laying simulation, the model of the powder laying device established by it not only affects the accuracy of the powder laying simulation, but also determines the difficulty of the simulation. The physical platform referenced by the simulation is composed of a powder recoater, a scraper, a powder supply cylinder, and a molding cylinder. The diameter of the powder supply cylinder reaches 150 mm, and the particle size of the powder is actually used between 25–106 μm. If the powder laying process model is established according to the ratio of 1:1, the completion of a powder layer thickness of 1 mm will require the generation of more than 400 million particles, which will undoubtedly make the simulation more difficult to process subsequent data. Therefore, in order to reduce the amount of calculation, this paper uses the sampling method. Only a small area of the powder laying process is selected for analysis, and the powder laying model is appropriately simplified. The simplified powder laying model is shown in Figure 5. The powder laying platform is reduced to a $20 \times 5.2 \times 1$ mm³ box. The molding cylinder and the powder supply cylinder are simplified to a cylinder with a diameter of 5 mm and a height of 4 mm. In order to ensure that the thickness of the powder laying layer is constant during simulation, the molding cylinder is in a mating state, and the distance from the powder laying platform is fixed at 0.2 mm; meanwhile, the powder supply cylinder

can move freely in the vertical direction to realize the powder supply function, which is floating.

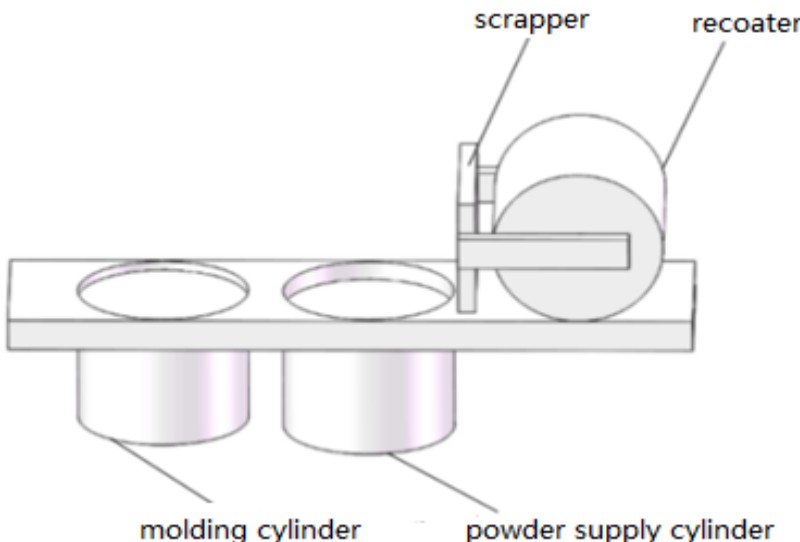

**Figure 5.** Model of the powder laying device.

In order to ensure the accuracy of the simulation results, the model of the powder particles is set according to the characteristics of the actual powder. The actual powder material referred to in this study is the 300–500 mesh spherical 316L stainless steel powder provided by Tuopu Metal Materials Co., Ltd., Chongqing, China. The loose density is 3.86 g/cm$^3$ and the spherical shape of the powder particles is satisfactory. The particle size distribution is uniform, which is guaranteed by Tuopu Metal Materials Co., Ltd, Chongqing, People's Republic of China. The spherical particles that follow the normal distribution, and the average particle size of 38 μm are selected in the simulation, and its microscopic morphology is shown in Figure 6.

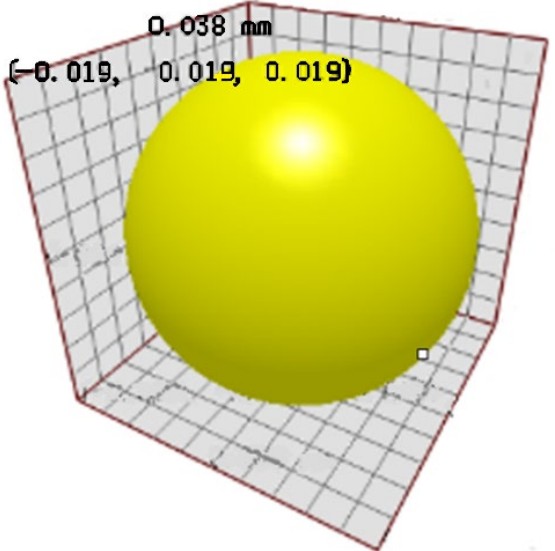

**Figure 6.** Microscopic morphology of powder particles.

### 4.1. Setting DEM Model Parameters

Before achieving the powder laying movement, the contact between the models and the material properties needs to be set. The powder particle material is 316L stainless steel, and the powder recoater is made of carbon steel. By consulting the DEM commonly

used material database and the friction coefficient table, the specific properties and mutual contact coefficients are imported as shown in Tables 1 and 2 [18,19].

**Table 1.** Material properties.

| Material | Poisson's Ratio | Density | Modulus of Rigidity |
|---|---|---|---|
| 316L Stainless Steel | 0.3 | 7.98 g/cm$^3$ | 75 Gpa |
| Carbon Steel | 0.27 | 7.89 g/cm$^3$ | 82 Gpa |

**Table 2.** Contact coefficients.

| Objects Being Contacted | Recovery after Collision | Static Friction | Rolling Friction |
|---|---|---|---|
| Particle–particle | 0.5 | 0.3 g/cm$^3$ | 0.05 |
| Particle–device | 0.3 | 0.15 g/cm$^3$ | 0.01 |

*4.2. Realization of the Powder Spreading Movement Process*

Figure 7 shows the simulated motion of the powder laying device. First of all, in order to make the way of powder particle generation similar to the actual material process, a virtual plane is set up above the powder supply cylinder as the formation surface of the particles, and a total of 300,000 particles are generated by dynamically generating particles into the cylinder to complete the filling of the powder supply cylinder. The powder supply cylinder is then raised so that the powder particles come to the work plane to achieve the supply of powder. Finally, the radius of the spreading recoater is adjusted and the translation velocity and angular velocity is set to push the powder from the powder supply cylinder to the molding cylinder. During the simulation process, since the simulation selects a small part of the entire powder laying process, in order to make it consistent with the particle flow state in the complete powder laying process, the periodic boundary conditions are set on both sides of the axial side of the powder laying platform, and in order to ensure that the powder supply is sufficient during the simulation, the height of the powder supply cylinder is set to two times the thickness of the powder laying layer. After the simulation is completed, the total mass M of the powder particles in the molding cylinder is calculated by meshing, and the density of the powder layer in the molding cylinder is calculated.

$$\rho = M/\pi r_a{}^2 h \tag{10}$$

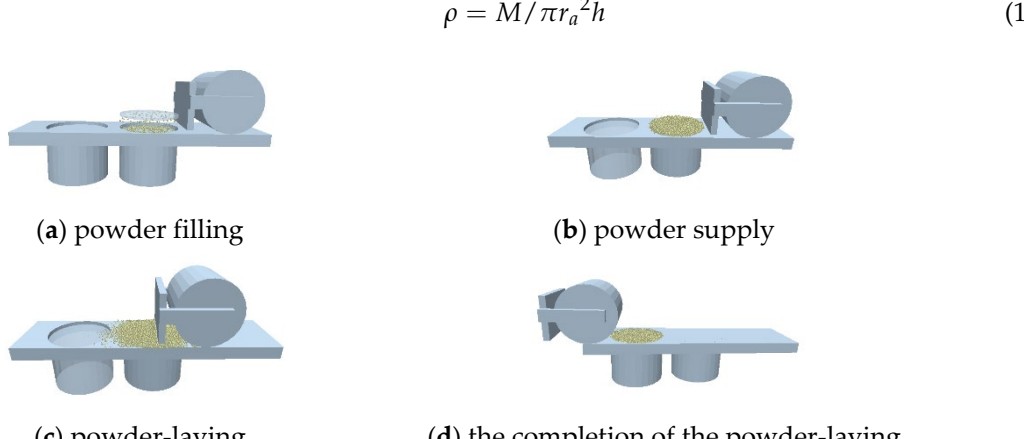

(**a**) powder filling

(**b**) powder supply

(**c**) powder-laying

(**d**) the completion of the powder-laying

**Figure 7.** Simulates the motion process of the powder laying.

In Equation (10), $r_a$ is the radius of the molding cylinder after its simplification, and $h$ is the distance between the molding cylinder and the powder laying platform.

**5. Simulation Results and Discussion**

Under the condition that the thickness of the coating layer is fixed to 0.2 mm, the simulation first uses the radius of the recoater $r$ = 15 mm, the translation velocity $v$ = 20 mm/s,

and the angular velocity $\omega = 8\,\text{s}^{-1}$ as a set of basic parameter settings, and the single factor method is used to study the influence of each parameter change on the density of the powder layer in turn. Then, according to the regular results, the appropriate parameter level is selected for orthogonal test, and the influence of each powder laying parameter on the density of the powder layer is discussed. Finally, the optimal parameter combination scheme is proposed.

### 5.1. The Influence of the Radius of the Recoater on the Quality of the Powder Layer

As one of the core components of the powder laying device, the radius size of the powder recoater directly affects the force direction of the powder particles when laying the powder, so as the radius of the powder recoater changes, the arrangement of the powder particles in the powder laying layer will inevitably change. Figure 8 shows the curve of the influence of the radius of the powder recoater on the density of the powder layer. Table 3 provides motion parameters and structural parameters of the recoater with different radiuses during its movement.

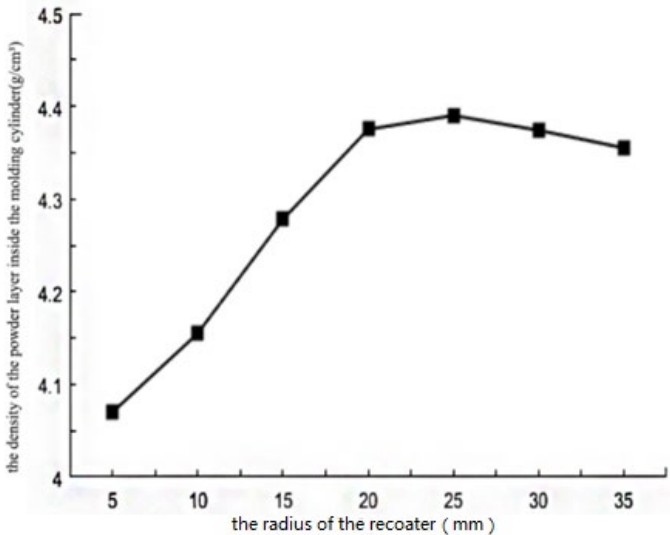

**Figure 8.** Variation curve of powder layer density under the radius of different recoaters.

**Table 3.** Motion parameters and structural parameters during the movement of the powder recoater with different radiuses.

| Radius (mm) | 5 | 10 | 15 | 20 | 25 | 30 | 35 |
|---|---|---|---|---|---|---|---|
| Angular velocity ($\text{s}^{-1}$) | | | | 8 | | | |
| Translation velocity (mm/s) | | | | 20 | | | |

As shown in Figure 8, as the radius of the powder recoater increases, the overall density of the powder layer shows a trend in which the overall desity of the powder layer first rises before falling. When the radius of the pawnshop powder recoater is less than 20 mm, the growth trend of the powder layer density is more obvious, because when the radius of the pawnshop powder recoater is small, the height of the powder layer being compacted is low, and as the radius increases, the height of the compaction increases; subsequently, the density of the powder layer is improved. When the radius of the powder recoater is greater than 20 mm, the growth of the powder layer density gradually tends to be flattened or even decreased, and the main reason for this is that when the radius of the powder recoater is too large, the height of the powder layer being compacted is close to the thickness of the powder supplied. At this time, the radius of the power recoater continues to increase and the compaction of the powder layer has basically no impact; however, it will make the kinetic energy carried by the powder recoater increase, resulting in the enhanced

mutual extrusion between the powder particles, thereby extruding the laid powder from the molding cylinder, which destroys the powder layer.

### 5.2. Effect of the Translation Velocity of the Powder Recoater on the Quality of the Powder Layer

The translation velocity of the powder recoater is an important indicator to measure the molding efficiency. In order to shorten the molding time, the translation velocity should be maintained at a high level, but the translational velocity of the powder recoater determines the horizontal distance of the powder following the movement of the powder recoater, and the change in its value, also affects the arrangement of the powder particles. The influence curve of the translation velocity of the powder recoater on the density of the powder layer is shown in Figure 9. Table 4 provides motion parameters and structural parameters of the recoater with different translation velocities during its movement.

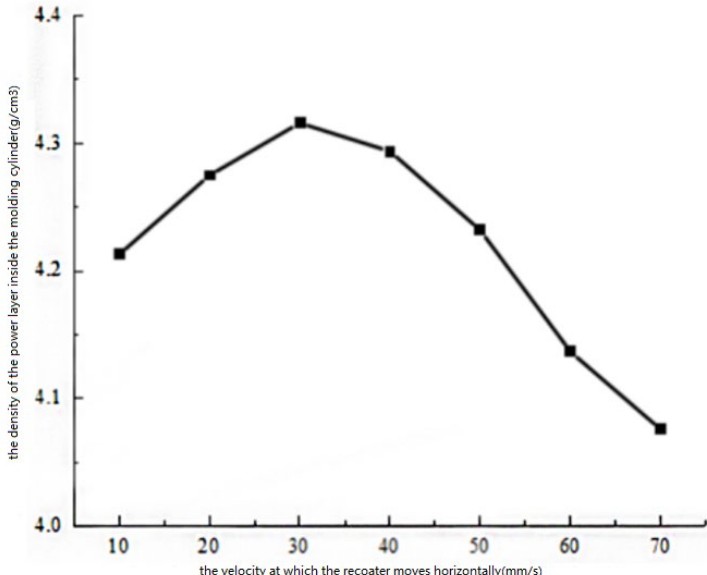

**Figure 9.** Variation curve of powder layer density under different translation velocities of powder recoaters.

**Table 4.** Motion parameters and structural parameters with different translation velocities during the movement of the powder recoaters.

| Radius (mm) | | | | 15 | | | |
|---|---|---|---|---|---|---|---|
| Angular velocity ($s^{-1}$) | | | | 8 | | | |
| Translation Velocity(mm/s) | 10 | 20 | 30 | 40 | 50 | 60 | 70 |

As show in Figure 9, under the condition that the radius and angular velocity of the powder recoater are constant, the translation velocity is increased, and the density of the powder layer shows a trend of increasing first and then decreasing. The main reason for this analysis is that, when the translation velocity of the pawnshop powder recoater is small, the displacement of the powder particles in the horizontal direction is small, and the gap between the powder particles cannot be filled; therefore, the density of the powder layer is low, and as the translation velocity increases, the flow of the powder is enhanced, the particle arrangement gap is gradually filled, and the density of the powder layer is improved. However, with the further increase in the translational velocity, the horizontal displacement distance of the powder particles per unit time is too long, resulting in some powder particles directly following the powder recoater and not falling into the molding cylinder, resulting in a decrease in the density of the powder layer.

### 5.3. Effect of the Angular Velocity of the Recoater on the Quality of the Powder Layer

The rotation of the powder recoater is conducive to improving the fluidity of the powder, and the size of its velocity directly affects the flow direction of the powder particles. Figure 10 shows the curve of the effect of the angular velocity of the powder recoater on the density of the powder layer. Table 5 provides motion parameters and structural parameters of the recoater with different angular velocities during its movement.

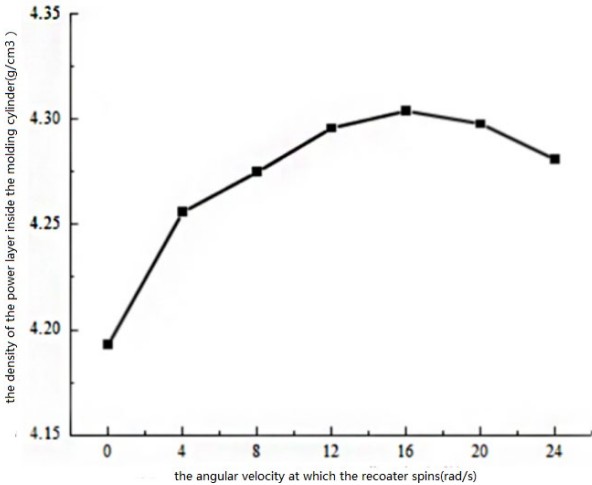

**Figure 10.** Variation curve of powder layer density under different angular velocities of powder recoaters.

**Table 5.** Motion parameters and structural parameters with different angular velocities during the movement of the powder recoater.

| Radius (mm) | | | | 15 | | | |
|---|---|---|---|---|---|---|---|
| Angular velocity (s$^{-1}$) | 0 | 4 | 8 | 12 | 16 | 20 | 24 |
| Translation velocity (mm/s) | | | | 20 | | | |

As show in Figure 10, in the process of the pawnshop powder recoater progressing from no self-rotation to angular velocity 16 s$^{-1}$, the density of the powder layer gradually increases, and the force of the powder particles in the oblique direction gradually increases in this process, indicating that, when the angular velocity is less than 16 s$^{-1}$, the force of the powder particles in the oblique direction makes its displacement ability enhanced, which is conducive to filling the gap between the large particles, especially when the powder recoater is transferred from no self-rotation to angular velocity 4 s$^{-1}$. The movement mode of the powder particles changes from pure flat to flat and rolling, and the lifting effect is more significant at this time. When the angular velocity of the pawnshop powder recoater is 16–24 s$^{-1}$, the density of the powder layer decreases with the increase in the angular velocity, and the force of the powder particles in the oblique direction continues to increase in this process, indicating that when the angular velocity is greater than 16 s$^{-1}$, the rotation of the powder recoater is too fast, resulting in a large oblique upward velocity of the powder particles, and the powder forms a splash without being compacted so that the density of the powder layer is reduced. When the angular velocity of the recoater is 12–20 s$^{-1}$, the change in the density of the powder layer is not obvious with the change of the angular velocity.

### 5.4. Optimization Analysis of Powder Laying Parameters

After obtaining the influence law of each parameter on the density of the powder layer by singular method, in order to accurately obtain the optimal combination of parameters of the powder laying, the orthogonal test method is used to select the three suitable horizontal values of the three parameters of the powder recoater radius *r*, translation velocity *v*, and

angular velocity $\omega$, and establish the factor level table as shown in Table 6. According to the horizontal table, the nine sets of orthogonal test protocols and the resulting data shown in Table 7 are obtained.

**Table 6.** Test factor levels.

| Levels | Factors | | |
|---|---|---|---|
| | *r* (mm) | *v* (mm/s) | $\omega$ (s$^{-1}$) |
| 1 | 20 | 20 | 12 |
| 2 | 25 | 30 | 16 |
| 3 | 30 | 40 | 20 |

**Table 7.** Orthogonal test protocol and result data.

| Number | *r* (mm) | *v* (mm/s) | $\omega$ (s$^{-1}$) | $\rho$ (g/cm$^3$) |
|---|---|---|---|---|
| 1 | 20 | 20 | 12 | 4.394 |
| 2 | 20 | 30 | 16 | 4.418 |
| 3 | 20 | 40 | 20 | 4.374 |
| 4 | 25 | 20 | 16 | 4.427 |
| 5 | 25 | 30 | 20 | 4.462 |
| 6 | 25 | 40 | 12 | 4.409 |
| 7 | 30 | 20 | 20 | 4.386 |
| 8 | 30 | 30 | 12 | 4.434 |
| 9 | 30 | 40 | 16 | 4.361 |

In order to analyze the significant influence of each powder laying parameter on the density of the powder layer, the result data is processed by the range analysis method, and the mean $k_i$ ($i$ = 1, 2, 3) of the test results under the same level of each factor is calculated separately. Then, the difference between the maximum value and the minimum value of the factor $k_i$ at different levels is obtained. The calculated results of the range analysis is shown in Table 8. According to Table 8, the ranges $R_v > R_r > R_\omega$; that is, the translation velocity of the powder recoater $v$ has the most significant influence on the density of the powder layer, the radius $r$ is second, and the angular velocity $\omega$ has the least impact. The higher the density of the coated layer, the better the corresponding powder laying effect; therefore the maximum value of $k_i$ under each factor should be taken as the optimal horizontal state; then the corresponding optimal parameter combination is the radius of the powder recoater 25 mm; the translation velocity is 30 mm/s, and the angular velocity is 12 s$^{-1}$, according to the optimized parameter combination. The powder laying verification was carried out on the basis of the original DEM model, and the density of the powder layer is 4.483 g/cm$^3$, which is higher than the maximum value in the 9 experimental protocols shown in Table 9. Therefore, the rationality of the parameter optimization scheme is verified theoretically.

**Table 8.** Results of the range analysis.

| Calculation Value | *r* (mm) | *v* (mm/s) | $\omega$ (s$^{-1}$) |
|---|---|---|---|
| $k_1$ | 4.395 | 4.402 | 4.412 |
| $k_2$ | 4.433 | 4.438 | 4.402 |
| $k_3$ | 4.394 | 4.382 | 4.407 |
| *R* | 0.039 | 0.056 | 0.010 |

**Table 9.** Pollination experiment result data.

| Number | *r* (mm) | *v* (mm/s) | $\omega$ (s$^{-1}$) | $\rho$ (g/cm$^3$) |
|---|---|---|---|---|
| 1 | 25 | 30 | 12 | 4.238 |
| 2 | 25 | 30 | 12 | 4.241 |
| 3 | 25 | 30 | 12 | 4.232 |

## 6. Powder Spreading Experiments Verified

In order to study the difference between the actual powder laying effect and the DEM simulation, the powder laying experiment was carried out on the self-developed laser molding equipment. Using 300–500 mesh spherical 316L stainless steel as the experimental material, adjusting the radius of the powder recoater to 25 mm, with the translational velocity of the scraper and the powder recoater at 30 mm/s, the angular velocity of the powder recoater at 12 s$^{-1}$, and the thickness of the powder laying layer at 0.2 mm, with the actual powder laying effect as shown in Figure 11, the powder layer surface is flat and without cracks, and the powder laying effect is satisfactory. This indicates that the optimized powder laying process parameter combination is required in the requirements of laser molding, and the substrate in the molding cylinder is removed after the powder laying is completed. The total mass m of the powder layer was measured using a precision electronic balance of 0.01 g, and the recorded data results are shown in Table 9.

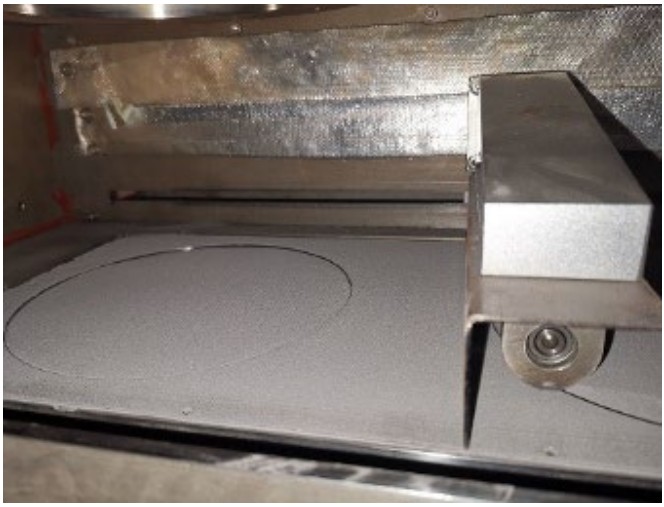

(**a**) Before the removal of the substrate

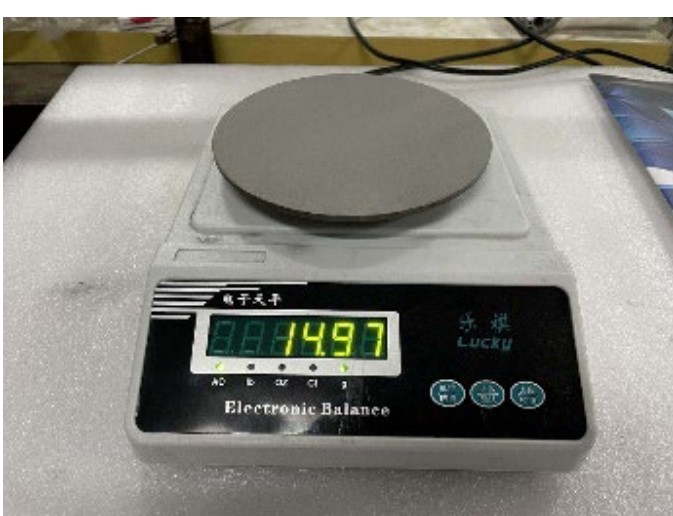

(**b**) After the removal of the substrate

**Figure 11.** Actual powder laying effect.

After three sets of experimental results were obtained, the average density of the powder layer was 4.237 g/cm$^3$, and the error between the simulation results and experiment results was 5.81%, which was within the acceptable range. The main reasons for the analysis error are: 1. When the powder layer measures the quality, the manual removal of the substrate causes the powder to be slightly spilled; 2. The simulation of the discrete unit cannot complete the simulation of the physical properties and interaction coefficients of the actual powder; 3. When the actual equipment is running, the movement process of the powder recoater has a slight vibration, which affects the density of the powder layer. In general, the experimental results are relatively close to the simulation results and the actual powder laying effect is satisfactory, which shows that the DEM powder laying model established in this paper is basically accurate, and explains the rationality of the parameter optimization scheme from another angle.

## 7. Conclusions

The DEM model of the laser powder bed fusion powder laying device was established, and the influence of the radius size, translation velocity, and angular velocity of the powder recoater on the density of the powder layer were systematically analyzed. The conclusions are drawn as follows: (1) Increased the radius of the powder recoater could increase the density of the powder layer within a certain range, and the influence of continuous improvement on the density of the powder layer was reduced. (2) When the translation velocity and angular velocity of the powder recoater is increased, the density of the powder layer showed a trend of first increasing and then decreasing, but the impact of the angular

velocity on the density of the powder layer was relatively small. (3) Through the verification of the powder laying experiment, the optimal combination of process parameters was obtained as follows: the radius of the powder recoater was 25 mm, the translation velocity was 30 mm/s, and the angular velocity was 12 s$^{-1}$. At this time, the surface of the powder layer was flat and had no cracks, and the powder laying effect was satisfactory, which was required in the requirements of laser molding.

The recommended data, which were analyzed in this study, are only applicable to specific powder and specific equipment, when other types of equipment and powder are employed, the method of this article can be referred to for other research to obtain suitable equipment parameters and achieve a high-quality powder coating effect.

**Author Contributions:** Conceptualization, Y.S. and G.L.; methodology, J.W. and K.Z.; software; Y.S., G.L. and K.Z.; validation, J.W. and G.L.; formal analysis, Y.S., G.L. and K.Z.; investigation, Y.S. and J.W.; resources, J.W.; data curation, Y.S. and K.Z.; writing—original draft preparation, Y.S. and G.L.; writing—review and editing, J.W. and K.Z.; visualization, Y.S. and G.L.; supervision, J.W.; project administration, J.W.; funding acquisition, J.W. All authors have read and agreed to the published version of the manuscript.

**Funding:** This research received no external funding.

**Institutional Review Board Statement:** Not applicable.

**Informed Consent Statement:** Not applicable.

**Data Availability Statement:** The study did not report any data.

**Conflicts of Interest:** The authors declare no conflict of interest.

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
