# Peer review of "Simulation for Discrete Elements of the Powder Laying System in Laser Powder Bed Fusion"

_metals, doi:10.3390/met12081375_

Round 1

Reviewer 1 Report

The optimization and simulation of different aspects of the PBF-LB/M process find (arouse) greater and greater curiosity within the researchers.
Therefore, the proposed topic is interesting and valuable for potential readers. However, the paper needs some revision and corrections:

Please, use the uniform standard definitions of AM processes described in ASTM ISO/ASTM52900-21 "Additive manufacturing — General principles — Fundamentals and vocabulary". There is no need to produce new names in AM industry. However, if some of the features are not described in the standard, please, check the literature. For example, "laying roller" is minor in the published literature, and more regular is "recoater".

L65, p. 2 - please add the number of references immediately after the researcher's surname to avoid readers' confusion. For example, "Yang, P. [12]".
Please elaborate on what "mortar laying device" means or change it to the appropriate word.

L185, p. 6 - it seems that the preposition "with" is missed in the sentence
Figure 5 - please use the appropriate for AM industry definitions

L244, p. 8 - what does "uniform" size mean? Please prove it by tests.

Conclusions. Please add that the proposed parameters are suitable only for specific (analyzed in this research) powder and particular machine setup. In different conditions, the proposed model will not be accurate.

Reviewer 2 Report

The main drawback of this paper is that simulation results differ from experimental data by about 5.81%. This value is bigger than the percentage that can be estimated from R´s values y average of each factor obtained from simulation, Table 8. Also, experimental validation as presented is not conclusive, i.e., it is not shown if simulation tendencies follow at least qualitatively the experimental ones, because only one set of experimental values was tested.

The title and abstract of the article must be improved to give a precise idea about the matter that it is dealing with. Some english improvements are needed to clarify ideas.

Reviewer 3 Report

The article clearly describes the definition and issues of Selective Laser Melting (SLM). The authors correctly state that it is an additive technology. The article describes the model and its simulation. The authors clearly describe the method of motion and the associated modelling.

The strength of the article is a clearly done research and an overview of who and what has been achieved in the discussed issue. A theoretical and mathematical model is constructed to address the issue. In my opinion, the strength of the paper is the application of the chosen methods in the paper and thus the results obtained. These results then provide a variety of interesting information from the simulations for their subsequent use. The thesis is clearly written.

On the other hand, the weak side is the large number of both formal and factual errors in formulas, units, statements in the text, tables, etc. Below is a full list of the errors and shortcomings found:

-       - a space is missing in the text between the numeric value and the unit in some cases, indicated in lines 223, 230, 232, 234, 245, 308, 312, 452 and 453;

-    -  for angular velocity, the unit rad/s is usually not written, but only s-1. The value of the angular velocity with the unit rad/s is incorrectly given in rows 358, 360, 363, 365, 368, 371, 401, 415, 453 and in Tables 3, 4, 6, 7, 8 and 9;

-     -   density is in the text with the unit g/cm3, this unit is missing in the header in Table 1 for Density. Also in the text (lines 403 and 431) the 3rd power is not in the superscript but part of the text. Wrong is g/cm3, correct is g/cm3;

-       - all formulas have a font size significantly larger than the rest of the text;

-       - in formulae 4, 5 and 6 there are discrepancies in the expression of the quantities.  In formula 4, the normal contact force is FCN,D , which does not correspond to the quantity in formula 6. Similarly, the Hertz contact constant in formula 5 (- Kn vs. Kn) does not correspond. The labels must be aligned and correspond in the formulas, otherwise it may not be clear exactly what is meant by the quantities;

-      -  Fig 6a has a completely illegible scale and in the case of Fig 6b the dimensions are not clear on the digital model, there are different numbers around the sphere, it is not clear which is the average or the deviation, or what the numbers mean;

-    -    another problem are the units in the tables. There is a lack of consistency in the units either for the name of the quantity in the table header or for the numerical values. This inconsistency is in Table 1 (Modulus of Rigidity), Table 3 (Rotational Speed, Translation Speed), Table 4 (Radius, Rotational Speed), Table 5 (Radius, Translation Speed). In the cases of Tables 6 and 7, the quantity is either above the column of numbers or next to it - this must also be made consistent, then it will be clear in the tables to which numbers the quantities belong;

-       - the text in subsection 5.3 is in bold, which is incorrect (lines 348-351), the word "effect" in the title of this subsection is lowercase, the correct word should be "Effect ..."

-     -   the titles of some subchapters are in bold, which is also wrong; these are the titles of subchapters 3.2 (line 183) and 5.1 (line 296);

-     -   the designation of Chapter 6 is on line 408 and Conclusions on line 441, (Conclusion should therefore be Chapter 7).

In conclusion, I positively evaluate the overall summary in three points. However, it is worth mentioning the practical application of the results. In particular, where can the results of the simulations be used and what is their significance or benefit in industry, production or even further development.

Round 2

Reviewer 1 Report

The authors have revised the manuscript entitled "Simulation for the Influence Factors of the Density of the Powder Layer in Selective Laser Melting" well. Paper may be accepted in its current form.
